# Clinically Relevant Chemotherapeutics Have the Ability to Induce Immunogenic Cell Death in Non-Small Cell Lung Cancer

**DOI:** 10.3390/cells9061474

**Published:** 2020-06-16

**Authors:** Tal Flieswasser, Jinthe Van Loenhout, Laurie Freire Boullosa, Astrid Van den Eynde, Jorrit De Waele, Jonas Van Audenaerde, Filip Lardon, Evelien Smits, Patrick Pauwels, Julie Jacobs

**Affiliations:** 1Center for Oncological Research (CORE), Integrated Personalized and Precision Oncology Network (IPPON), 2610 Wilrijk, Belgium; jinthe.vanloenhout@uantwerpen.be (J.V.L.); laurie.freireboullosa@uantwerpen.be (L.F.B.); astrid.vandeneynde@uantwerpen.be (A.V.d.E.); jorrit.dewaele@uantwerpen.be (J.D.W.); jonas.vanaudenaerde@uantwerpen.be (J.V.A.); filip.lardon@uantwerpen.be (F.L.); evelien.smits@uza.be (E.S.); patrick.pauwels@uza.be (P.P.); julie.jacobs@uantwerpen.be (J.J.); 2Department of Pathology, Antwerp University Hospital, 2650 Edegem, Belgium; 3Center for Cell Therapy and Regenerative Medicine, Antwerp University Hospital, 2650 Edegem, Belgium

**Keywords:** non-small cell lung cancer, chemotherapy, immunogenic cell death

## Abstract

The concept of immunogenic cell death (ICD) has emerged as a cornerstone of therapy-induced anti-tumor immunity. To this end, the following chemotherapies were evaluated for their ability to induce ICD in non-small cell lung cancer (NSCLC) cell lines: docetaxel, carboplatin, cisplatin, oxaliplatin and mafosfamide. The ICD hallmarks ATP, ecto-calreticulin, HMGB1, phagocytosis and maturation status of dendritic cells (DCs) were assessed in vitro. Furthermore, an in vivo vaccination assay on C57BL/6J mice was performed to validate our in vitro results. Docetaxel and the combination of docetaxel with carboplatin or cisplatin demonstrated the highest levels of ATP, ecto-calreticulin and HMGB1 in three out of four NSCLC cell lines. In addition, these regimens resulted in phagocytosis of treated NSCLC cells and maturation of DCs. Along similar lines, all mice vaccinated with NSCLC cells treated with docetaxel and cisplatin remained tumor-free after challenge. However, this was not the case for docetaxel, despite its induction of the ICD-related molecules in vitro, as it failed to reject tumor growth at the challenge site in 60% of the mice. Moreover, our in vitro and in vivo data show the inability of oxaliplatin to induce ICD in NSCLC cells. Overall with this study we demonstrate that clinically relevant chemotherapeutic regimens in NSCLC patients have the ability to induce ICD.

## 1. Introduction

Non-small cell lung cancer (NSCLC) represents an estimated 85% of all lung cancers, accounting for 2.1 million new lung cancer cases and approximately 1.8 million deaths per year worldwide [1]. It remains the leading cause of cancer mortality worldwide with a 5-year overall survival rate of only 15% for all stages, according to the World Health Organization [2,3]. The first-line treatment of advanced NSCLC in the majority of patients still consists of conventional chemotherapy (regimens with platinum-based agents) to achieve tumor response or stable disease [4].

In the last decade, important therapeutic advances took place in the treatment of NSCLC, such as development of small molecular tyrosine kinase inhibitors (TKIs) targeting specific genetic alterations as well as immunotherapy, which has paved its way into the clinic as first- and second-line treatment [5]. Especially the emergence of immune checkpoint inhibitors, more specifically anti-programmed cell death protein 1 (PD-(L)1), represents a landmark of success in a broad range of tumor types with a great number of ongoing clinical trials and recent Food and Drug Administration (FDA) and European Medicines Agency (EMA) approvals to treat different tumor types, including NSCLC [6,7,8,9,10,11]. Although recent targeted and immunotherapeutic strategies have shown remarkable clinical responses as single agents, such durable responses are only observed in a minority of patients and in addition, resistance nearly always develops [12,13]. Thus going beyond monotherapy to combination regimens which increase immunostimulatory effects is a worthwhile strategy to circumvent this challenge [14].

While in the past, conventional cancer chemotherapeutic agents were used with the main aim of achieving tumor cell toxicity, recent studies have demonstrated that certain chemotherapeutic drugs can modulate the anti-tumor immune response [15,16]. Different conventional chemotherapeutic agents can activate innate and adaptive anti-tumor responses, and thus increase treatment efficacy [17]. The immunomodulatory properties of chemotherapeutic agents, commonly used for NSCLC in the clinic, have already been described in literature to some extent. For instance, cisplatin (CDDP) has been shown to reduce the number of myeloid-derived suppressor cells in a murine B16 melanoma model [18], while another study reported that CDDP can sensitize tumor cells to Fas-mediated tumor cell killing by enhancing Fas expression on tumor cells, thereby increasing their vulnerability to immune cells expressing Fas ligand [19]. In addition, CDDP has been shown to upregulate expression of MHC class I chain related molecule A and B, leading to enhanced NK cell-mediated killing of NSCLC cells [20,21]. Moreover, CDDP and its analog, carboplatin (CARBO), can induce activation of macrophages and increase their antitumor activity against murine sarcoma cells [22]. Furthermore, docetaxel (DOC) treatment of peripheral blood samples of NSCLC patients resulted in a reduced amount of regulatory T cells in vitro [23].

In addition to these immunomodulatory properties of chemotherapeutic agents, another way to efficiently activate a long-term anti-tumor immune response is through the induction of immunogenic cell death (ICD) [24]. Therapy-treated tumor cells succumbing to ICD release certain damage-associated molecular patterns (DAMPs), such as early lysosomal adenosine triphosphate (ATP) secretion (find-me signal), pre-apoptotic translocation of the endoplasmic reticulum (ER) chaperone protein calreticulin (ecto-CALR) to the tumor cell surface (eat-me signal) and post-mortem release of the nuclear protein high mobility group box 1 (HMGB1) [16,25,26]. Upon release of DAMPs, dendritic cells (DCs) are recruited to the tumor site. There, they process tumor antigens and undergo subsequent maturation and activation, leading to activation of an adaptive anti-tumor immune response [27]. The ‘gold standard’ to confirm the ability of a therapeutic agent to ICD, currently relies on the in vivo vaccination assay [28]. In this setting, immune competent mice are subcutaneously (s.c.) injected with the vaccine (treated tumor cells) on one side of the flank, while the challenge (live cells) is injected in the opposite side of the flank. The percentage of tumor-free mice represents the degree of immunogenicity of the tested compound [28]. To date, chemotherapies that are mostly used as positive controls for ICD include oxaliplatin (OXA), cyclophosphamide (CP), mitoxantrone, epirubicin and doxorubicin to compare the immunogenicity of a therapeutic agent in certain cancer models [28,29,30,31,32,33].

Currently, extensive data on the ICD potential of clinically relevant chemotherapeutics for NSCLC is lacking. To this end our data provide novel insights into the in vitro and in vivo potential of various first- and second-line chemotherapeutic agents for NSCLC using a panel of human and murine NSCLC cell lines.

## 2. Materials and Methods

### 2.1. Cell Lines and Cell Culture

Cell lines A549, NCI-H1975 and NCI-H1650 were purchased from the American Type Culture Collection (ATCC, Molsheim Cedex, France). In addition, the Lewis Lung carcinoma (3LL) cell line (gift from Dr. Carsten Riether, Department of Clinical Research, University of Bern; derived from the lung of a C57BL/6J mouse) was included. A549 and 3LL were cultured in Dulbecco’s Modified Eagle Medium (DMEM, Life Technologies, 10938-025, Merelbeke, Belgium) supplemented with 10% FBS (Life Technologies, 10270-106), 1% penicillin (100 U/mL) /streptomycin (100 µg/mL; Life Technologies, 15140-122) and 2 mM l-glutamine (l-Glut, Life Technologies, 25030-024). NCI-H1975 and NCI-H1650 were cultured in Roswell Park Memorial Institute Medium (RPMI, Life Technologies, 52400-025) supplemented as described above. Cells were grown as monolayers and were maintained in exponential growth in 5% CO_2_ + 95% air in a humidified incubator at 37 °C. All cell cultures were confirmed as Mycoplasma free using the Mycoalert^®^ Mycoplasma detection kit (Lonza, LT07-218, Verviers, Belgium).

### 2.2. Cytotoxicity of Different Chemotherapies 

The following chemotherapeutic agents were used: DOC (S1148), CARBO (S1215), CDDP (S1166), OXA (S1224) and pemetrexed (PEM, S1135; all purchased from Selleckchem, Munich, Germany) and mafosfamide (MF, the active metabolite of cyclophosphamide; purchased from Niomech—IIT GmbH, D-17272). To evaluate cytotoxicity of the different chemotherapeutic agents, NSCLC cells were seeded in 96-well plates, incubated overnight and treated for 72 h with either CDDP (0–20 µM), DOC (0–50 µM), OXA (0–50 µM), MF (0–300 µM), CARBO (0–50 µM) or PEM (0–10 µM) as single agents. After treatment, cell monolayers were fixed with 10% trichloroacetic acid for 1 h at 4 °C and stained with 100 μL 0.1% sulforhodamine B, as previously described [34]. The inhibitory concentration (IC) of the drug that leads to 40% (IC_40_) and 50% (IC_50_) growth inhibition, was calculated using the WinNonlin software (Pharsight). All experiments were performed at least in triplicate. Combination regimens consisted of the IC_50_-72h value of DOC with the IC_40_ value of either CARBO or CDDP.

### 2.3. ATP Release

Extracellular ATP was measured in the conditioned media (supplemented with heat inactivated FBS) 24 h after chemotherapeutic treatment using the ENLITEN^®^ ATP assay system, according to the manufacturer’s protocol (Promega, FF2000, Leiden, The Netherlands). The bioluminescent signal was measured using a VICTOR^™^ plate reader (PerkinElmer).

### 2.4. Ecto-CALR Expression

Forty-eight hours after treatment, cells were harvested and incubated with 5% normal goat serum (NGS, Sigma-Aldrich, G9023), followed by washing and incubation with Alexa Fluor 488-conjugated anti-CALR (Abcam, ab196158, 1/100 dilution) antibody for 40 min. Prior to sample analysis, cells were stained with Annexin V (AnnV, APC-conjugated; BD, 550474) and propidium iodide (PI; BD Biosciences, 556463) to gate on non-permeabilized cells (AnnV^+/−^, PI^−^) for ecto-CALR expression, as previously described [35]. Isotype control (rabbit IgG, Abcam, 199091, 1/100 dilution, Cambridge, UK) was included to correct for non-specific binding. Flow cytometric acquisition was performed on a BD Accuri^™^ C6 instrument (BD Biosciences). Data analysis was performed using FlowJo v10.1 software (TreeStar).

### 2.5. HMGB1 Release

Release of HMGB1 was analyzed 72 h after treatment in the conditioned media using an enzyme-linked immunosorbent assay (IBL, ST51011) according to the instructions of the manufacturer. The absorption was measured via the iMARK^™^ plate reader (Bio-rad).

### 2.6. In Vitro Generation of Human Monocyte-Derived DCs

Buffy coats from healthy donors (Red Cross Flanders Blood Service, Belgium) were collected and human peripheral blood mononuclear cells (PBMCs) were isolated by LymphoPrep gradient separation (Sanbio, 1114547). Monocytes were isolated from PBMCs using CD14 microbeads according to the manufacturer’s protocol (Miltenyi Biotec, 272-01, Utrecht, The Netherlands) with purity >90% after isolation. Subsequently, cluster of differentiation (CD) 14 positive (^+^) cells were plated at a density of 1.25–1.35 × 10^6^ cells/mL in 1640 RPMI (Life Technologies, 52400-025) supplemented with 2.5% human AB (hAB, Sanbio, A25761) serum, 800 U/mL granulocyte-macrophage colony stimulating factor (GM-CSF; Gentaur, 04-RHUGM-CSF) and 20 ng/mL interleukin (IL)-4 (Miltenyi Biotec, 130-094-117) at day 0, as previously described [36].

### 2.7. Phagocytosis and Maturation Status of DCs

After in vitro generation of DCs, NCI-H1975, A549 and NCI-H1650 cells were labelled with PKH67, a green fluorescent membrane dye (Sigma Aldrich, MIDI67), prior to plating them out on day 3. NSCLC cells were treated with chemotherapy on day 4. On day 5, immature DCs were stained with cytoplasmic violet-fluorescent CellTracker Violet BMQC dye (Invitrogen, C10094) and effector (E) and target (T) cells were placed in coculture at a 1:1 (E:T) ratio. Supernatant (SN) was stored (−20 °C), cells were collected and immediately used for flowcytometric detection of DC maturation markers and phagocytosis on day 7. Cells were stained with CD80-PerCP5.5 (Biolegend, 400150) and CD86-PE-Cy7 (BD Biosciences, 557872) to assess DC maturation (Violet^+^ population). Isotype controls (PerCP5.5, Biolegend, 305232; PE-Cy7, BD Biosciences, 557872) were included to subtract aspecific signals from measured fluorescence intensity. Phagocytosis of NSCLC cells was assessed by gating on the PKH67^+^Violet^+^ population, as previously described [35]. Acquisition was performed on a FACSAria II (BD Biosciences). Data analysis was performed using FlowJo v10.1 software (TreeStar).

### 2.8. In Vivo Vaccination Assay

Six-week old immunocompetent female C57BL/6J mice were purchased from Charles River Laboratories and housed at the Animal Center (University of Antwerp) in randomized cages (five per cage) with a 12 h day/night cycle and with food and water ad libitum. 

#### 2.8.1. In Vitro Preparation of Treated NSCLC Cell Vaccine

On day 0, 3LL cells (1 × 10^6^ cells/mL) were plated out in a T175 flask (5 × 10^6^ cells in total) and resuspended in supplemented DMEM. On day 1, 3LL cells were treated with either vehicle (phosphate-buffered saline; PBS) or chemotherapy (IC_50_-72h for monotherapies and additional IC_40_-72h of either CARBO or CDDP in the combination regimen with DOC). On day 2, floating cells were collected, washed with PBS and adjusted to the appropriate concentration (1 × 10^6^ cells/100 µL). Washed 3LL cells were resuspended in PBS (10 × 10^6^/mL) and incubated (37 °C) for 48 h to decrease viability (viability varied around 26–61% at day 3) and guarantee no tumor growth at the vaccination site. On day 5, the vaccine of treated 3LL cells (viability below 10%) was subcutaneously (s.c.) injected at the lower-right side of the abdomen (100 µL/mouse). The following week, mice were vaccinated again according to the protocol, as described above.

#### 2.8.2. In Vitro Preparation of Live NSCLC Cells for Challenge

Live 3LL cells were collected and counted (with trypan blue) to adjust the concentration of living NSCLC cell suspension to 50,000 cells/100 µL of PBS per mouse. Live cells were initially injected seven days after 2nd vaccination at the challenge site on the left side of the abdomen. After a follow-up period of two weeks, all mice were re-challenged with live 3LL cells.

#### 2.8.3. In Vivo Follow-Up of Mice

Tumor size was measured three times a week using a caliper and weight was measured to detect potential toxicities. Mice were euthanized when reaching their ethical endpoint (volume > 1500 mm^3^ or deep ulceration). All animal experiments were approved by the Antwerp Ethics Committee for Animal Testing (ECD-2018-53).

## 3. Statistical Analysis

Prism 8.02 software (GraphPad) was used for data comparison and graphical data representations. SPSS Statistics 25 software (IBM) was used for statistical computations. The Kruskal-Wallis test was used to compare means between more than two groups. The Mann-Whitney U test was used to compare means between two groups. All statistical tests were performed on a minimum of three independent experiments. The Log-rank test was used to compare survival probability between different groups. *p*-value ≤ 0.05 was considered statistically significant.

## 4. Results

### 4.1. Cytotoxicity of Different Chemotherapeutic Agents towards NSCLC Cell Lines

Dose response curves of chemotherapeutic agents towards NSCLC cell lines were assessed in a panel of five NSCLC cell lines (Figure 1). Of note, PEM, a chemotherapeutic agent which is commonly used in NSCLC, was initially included in the experiments. However, no dose-response curve with cell viability of <50% could be obtained for all cell lines using PEM (0–10 µM; Appendix A).

While NCI-H1975 was most sensitive to DOC (0.029 µM ± 0.48), this cell line was most resistant to CDDP (20 µM ± 1.6) and OXA (42 µM ± 0.51) compared to other cell lines. NCI-H1650 cells were most resistant to CARBO (189.30 µM ± 1.38). The murine 3LL was most sensitive to CARBO (17 µM ± 2.20), CDDP (1.92 µM ± 1.74), OXA (1.25 µM ± 1.56) and MF (4.55 µM ± 2.65). Dose-response curves of A549 cells were mostly in-between other NSCLC cell lines. The cells were most resistant to MF (17.86 µM ± 1.32) and most sensitive to OXA (3.08 µM ± 0.56). Overall, these data demonstrate divergent sensitivity of NSCLC cell lines to different chemotherapeutic agents. Therefore, in order to compare the immunogenic potential of the chemotherapeutic agent in our panel of NSCLC cell lines, the different IC50-72h values per cell line were used in further experiments. Because we determined the IC50 values over a timeframe of 72 h of treatment, the sequential order of the DAMPs was evaluated over this period of time.

### 4.2. Release of DAMPs by NSCLC Cell Lines after Chemotherapy

#### 4.2.1. ATP Secretion

First, ATP secretion was assessed 24 h after treatment with chemotherapy in all NSCLC cell lines (Figure 2).

In the NCI-H1975 cell line treatment with all chemotherapies showed a significant 2-fold increase of ATP secretion compared to vehicle, except for treatment with CARBO. A549 cells treated with DOC, CARBO, MF and the two combination regimens showed a 2- to 3-fold significant increase of ATP compared to vehicle, with exception of CDDP and OXA. In NCI-H1650 cells, ATP levels were significantly increased after treatment with DOC, MF and the combination of DOC + CARBO by 2- to 4-fold compared to vehicle. Along the same line, murine 3LL cells treated with DOC, MF and the combination regimens showed a significant 2-fold increase of ATP secretion.

Overall, in all NSCLC cells lines, treatment with DOC, MF and DOC + CARBO induced significantly higher levels of ATP compared to vehicle. In addition, three out of the four NSCLC cell lines treated with DOC + CDDP resulted in a significant higher release of ATP compared to vehicle. However, no significant differences were found between the different chemotherapies.

#### 4.2.2. Ecto-CALR Exposure

Next, ecto-CALR exposure on NSCLC cells was assessed after 48 h of treatment with chemotherapy in all four NSCLC cell lines (Figure 3, Appendix A). For this, NSCLC cell staining was performed with AnnV/PI to gate on non-permeabilized cells (Appendix A). In NCI-H1975 cells, treatment with all chemotherapeutic agents significantly increased percentages of ecto-CALR positive cells compared to vehicle, ranging from 1% up to 8% (Figure 3). In the A549 cell line treatment with DOC, DOC + CARBO and DOC + CDDP significantly increased ecto-CALR positive cells compared to vehicle, although this increase was less pronounced compared to other cell lines. Similar to NCI-H1975, all chemotherapies significantly increased ecto-CALR positive cells in the NCI-H1650 cell line compared to vehicle, with exception of MF. In addition, a more pronounced increase of ecto-CALR positive cells was observed in murine 3LL cells, which significantly increased ecto-CALR positive cells after treatment with all chemotherapies except for OXA, ranging from 10% up to 40% of ecto-CALR positive cells compared to vehicle.

Overall, DOC, as monotherapy or in combination regimens, significantly increased ecto-CALR positive cells in all NSCLC cell lines. Moreover, treatment with DOC + CDDP showed higher %ecto-CALR positive cells compared to treatment with DOC and DOC + CARBO in the NCI-H1675 cell line (*p* ≤ 0.05). No significant differences between treatment with DOC, DOC + CARBO and DOC + CDDP were found in the other NSCLC cell lines.

#### 4.2.3. HMGB1 Release

Finally HMGB1 release was assessed after 72 h of treatment with chemotherapy in all four NSCLC cell lines (Figure 4). In the NCI-H1975 cell line, HMGB1 release was significantly increased compared to vehicle after treatment with DOC, DOC + CARBO and DOC + CDDP, with the latter reaching a nearly 4-fold increase compared to vehicle. Both combination strategies showed significantly higher amounts of HMGB1 compared to treatment with DOC (*p* ≤ 0.05). Similarly, A549 cells treated with DOC, DOC + CARBO and DOC + CDDP significantly increased HMGB1 release. Both combinations resulted in significantly higher levels of HMGB1 compared to treatment with DOC (*p* ≤ 0.05). In NCI-H1650 cells, only treatment with DOC + CARBO and DOC + CDDP showed a significant 2-fold increase of HMGB1, both at similar levels. Moreover, 3LL cells showed a significant 2- to 4-fold increase of HMGB1 release after treatment with all chemotherapies compared to vehicle, with exception of OXA.

Overall, in all NSCLC cell lines, both DOC + CARBO and DOC + CDDP significantly increased HMGB1 levels compared to vehicle. 

### 4.3. DC Phagocytic Function and Maturation Status after Chemotherapy Treatment of NSCLC Cell Lines

Phagocytosis of dying tumor cells and subsequent maturation of DCs is a crucial step in the process of establishing an adequate anti-tumor immune response [37]. To this end, human NCI-H1975, A549 and NCI-H1650 cells were treated with the different chemotherapies and were subsequently co-cultured with immature DCs of three healthy donors in order to assess their phagocytotic ability and maturation status (CD80 and CD86). First, potential cytotoxic effects of the chemotherapeutic agents on DC viability were evaluated. Treated DCs showed no difference in viability compared to their untreated (PBS) counterparts (Appendix A). Furthermore, co-cultures of NSCLC cells and DCs resulted in phagocytosis of treated NSCLC cells at varying levels depending on the cell line and on the chemotherapy (Figure 5). Overall treatment with DOC, DOC + CARBO and DOC + CDDP resulted in phagocytosis of NSCLC cells by DCs in all NSCLC cell lines.

In addition, maturation markers were upregulated on DCs after chemotherapy treatment at varying levels with CD86 expression being overall more pronounced than CD80 (shown as ΔMFI). Importantly, expression of CD80 and CD86 on monocultured DCs treated with the chemotherapeutic agents did not significantly differ from vehicle (Appendix A), indicating that the immunogenic effects of chemotherapy rely on their interaction with tumor cells. In the NCI-H1975 cell line, CD80 was significantly upregulated on DCs after treatment with CDDP and OXA. Similarly, CD86 was upregulated after CDDP treatment as opposed to the latter. Moreover, DOC, CARBO and the two combinations significantly upregulated CD86 on DCs in co-culture with NCI-H1975 cells. While DCs in co-culture with A549 cells only showed an upregulation of CD80 after treatment with DOC and CARBO, CD86 expression was significantly upregulated on DCs after treatment with all chemotherapies compared to vehicle. There were no statistical significant differences found in CD86 expression on DCs between the different chemotherapies. Co-cultures of DCs with NCI-H1650 cells showed a more pronounced upregulation of CD80 on DCs compared to the other cell lines. In the NCI-H1650 cell line, CD80 was significantly upregulated on DCs after treatment with CARBO, OXA, MF and both combination regimens compared to vehicle. Maturation marker CD86 was significantly upregulated after treatment with all chemotherapies, except for MF.

Overall, in all NSCLC cell lines upregulation of CD86 was more pronounced compared to CD80 and significantly upregulated after treatment with DOC, CARBO, CDDP, DOC + CARBO and DOC + CDDP compared to vehicle.

Since immunosuppressive cytokines, such as transforming growth factor-β (TGF-β) are largely produced within the tumor microenvironment, its secretion in co-cultures was measured and shown as foldchange relative to control (concentrations ranging from 300–3000 pg/mL; Suppl. Methods). In the co-culture with NCI-H1975 cells, TGF-β levels were significantly lower after treatment with DOC, CARBO, OXA and DOC + CARBO compared to vehicle. Similarly, co-cultures with A549 cells showed significantly lower levels of TGF-β after treatment with DOC, CARBO, OXA, DOC + CARBO as well as DOC + CDDP compared to vehicle. In addition, TGF-β levels were significantly lower compared to vehicle in co-cultures with NCI-H1650 cells after treatment with these above-mentioned chemotherapeutic regimens, with exception of CARBO.

In addition to TGF-β, secretion of immunostimulatory cytokines interferon-β (IFN-β), tumor necrosis factor-α (TNF-α) and IL-6 were assessed in the co-cultures. In case of IFNβ, its secretion in the SN of different treatment groups was not significantly higher compared to vehicle, except for co-cultures of NCI-H1975 cells treated with DOC + CDDP. Secretion of IL-6 was significantly higher in co-cultures of NCI-H1975 treated with each treatment regimen compared to vehicle, with exception of MF. In case of TNFα, both co-cultures with A549 and NCI-H1650 cells showed significantly higher levels after treatment with DOC and CDDP compared to vehicle. In addition, A549 cells treated with OXA and DOC + CDDP showed significantly higher levels of TNFα secretion compared to vehicle as well (Appendix A).

### 4.4. Overview of DAMPs and DC Phagocytic Function and Maturation Status after Chemotherapy Treatment of NSCLC Cell Lines

An overview of all data of in vitro assessment of DAMPs, phagocytosis and maturation of DCs in all NSCLC cell lines is depicted in Figure 6. Three out of four NSCLC cell lines (NCI-H1975, A549 and 3LL) showed significantly higher levels of ATP, ecto-CALR and HMGB1 after treatment with DOC, DOC + CARBO and DOC + CDDP compared to vehicle. In addition, phagocytosis of treated NSCLC cells and maturation (CD86) of DCs were significantly increased in all three human NSCLC cell lines after treatment with the above-mentioned chemotherapeutic regimens. Furthermore, murine 3LL cells treated with DOC, MF, DOC + CARBO and DOC + CDDP resulted in a significant release of all three DAMPs in vitro, as opposed to treatment with OXA. 

### 4.5. In Vivo Vaccination Assay in the 3LL Mouse Model

The in vitro experiments can give valuable insights into the potential of therapeutic agents to mount an effective anti-tumor immune response, although these assays do not take a complete immune system into account. Hence, we performed a vaccination experiment in C57BL/6J mice with murine 3LL cells in order to validate our in vitro results. Mice were vaccinated twice with (un)treated 3LL cells, subsequently challenged with live 3LL cells and then monitored for tumor growth during 60 days. All mice’s weight remained stable throughout the entire follow-up period. Percentage of tumor-free mice and follow-up of tumor growth are depicted in Figure 7.

Four out of five mice (80%) from the control group developed a tumor at the challenge site. Similarly, four out of five mice (80%) vaccinated with OXA-treated cells developed a tumor at the challenge site. Furthermore, three out of five mice (60%) vaccinated with DOC-treated cells developed a tumor at the challenge site. Mice vaccinated with CARBO, CDDP, MF and DOC + CARBO developed a tumor in only one out of five mice (20%). Notably, the tumor-bearing MF mouse failed to reach the endpoint of 1500 mm^3^ tumor volume due to early death (unknown cause). Furthermore, five out of five mice (100%) vaccinated with DOC + CDDP remained tumor-free during the entire follow-up period. A trend towards significant differences was observed between survival probabilities of the different groups (Log-rank test, *p* = 0.079).

## 5. Discussion

The ability of chemotherapy to elicit ICD has become a thoroughly investigated phenomenon, in both in vitro and in vivo settings [16,37]. To this end, we demonstrated that clinically relevant chemotherapeutic regimens for NSCLC can exert immunostimulatory effects in different NSCLC cell lines. Especially, DOC, DOC + CARBO and DOC + CDDP resulted in a significant increase of DAMPs and maturation of DCs in NSCLC cell lines in vitro. As opposed to DOC alone, combination regimens of DOC + CARBO and DOC + CDDP showed similar findings in an in vivo vaccination assay resulting in 80% and 100% tumor-free mice, respectively.

OXA, CARBO and CDDP all belong to the platinum-derived compounds and are cytotoxic by forming inter- and intra-strand DNA adducts [38]. Many articles performing research on the ability of chemotherapeutic agents to induce ICD report that CDDP is a negative control due to its inability to trigger ER stress, and hence its incapacity to promote CALR translocation from the ER lumen to the outer leaflet of the plasma membrane [39]. Contrarily, CDDP has also been described to exert pleiotropic effects, including induction of ER stress [40,41,42,43]. Ecto-CALR exposure on the membrane of (dying) tumor cells is important for recognition and engulfment by DCs, leading to optimal antigen presentation to T cells, hence activating an adaptive anti-tumor immune response [25]. Activation of the ER stress response is characterized by phosphorylation of the eukaryotic translation initiation factor 2α (eIF2α) by protein kinase-like ER kinase (PERK). Once activated, sequential activation of caspase-8 and cleavage of the ER protein (BAP31) is triggered, leading to CALR translocation from the ER to exposure on the cell surface [25].

One study showed that the IC_50_ of CDDP and OXA were both able to induce nuclear apoptosis and ATP release at similar levels. However, OXA induced higher phosphorylation of eIF2α and thus higher CALR levels compared to CDDP using U2OS cells, an osteosarcoma cell line. In addition, they showed that OXA-treated 3LL cells triggered higher ecto-CALR levels compared to CDDP-treated 3LL cells [39]. Contrary to their results, we observed significantly higher ecto-CALR levels in 3LL cells treated with CDDP compared to OXA-treated cells. While the study of Tesniere et al. treated cells with 300 µM of OXA and 150 µM of CDDP, we only used 1.25 µM and 1.92 µM (IC_50_ value determined as described in materials & methods), respectively, which might explain these discrepancies. Another possible explanation underlying these interstudy discrepancies is reported to be the production of IL-8 by various cancer cells. Sukkurwala et al. showed that cancer cells that produce high amounts of IL-8 in response to immunogenic chemotherapies stimulate CALR exposure [44]. In this regard, previous studies reported that certain cancer cells, including the NSCLC cell line A549, constitutively produce high levels of IL-8 [45,46], thus induction of ecto-CALR exposure in this cell line would only seem plausible. However, in our study A549 cells showed quite low %ecto-CALR positive cells after 48 h treatment with DOC, DOC + CARBO and DOC + CDDP. These interstudy discrepancies might be due to differences between concentrations used and/or duration of the treatment [47].

Another study showed that OXA, but not CDDP could trigger eco-CALR exposure using murine and human colon cancer cell lines, although both compounds resulted in HMGB1 release at similar levels [30]. Similarly, we observed that both CDDP- and OXA-treated NSCLC cells could trigger HMGB1 release. Moreover, both CDDP and OXA treatments showed phagocytosis of NSCLC cells by DCs and upregulation of maturation markers on DCs, although at varying levels among the different NSCLC cell lines. Our results are in line with previous studies showing increased tumor cell uptake and upregulation of maturation markers CD80 and CD86 on different subsets of human blood DCs after treatment with CDDP and OXA [48]. Since CDDP and OXA are thoroughly studied in the context of ICD, one would expect that treatment with other platinum analogs, such as CARBO could exert immunogenic effects on the tumor and its micro-environment as well. Indeed, in our study CARBO induced DAMPs and led to upregulation of maturation markers on DCs at varying levels.

To date, data on the ability of CARBO to induce ICD are quite limited. While one study reported that CARBO did not significantly increase ecto-CALR exposure and HMGB1 release in TSA cells (mouse mammary adenocarcinoma), another study reported its ability to induce ecto-CALR exposure and HMGB1 release in murine colon cancer cells [49,50].

Furthermore, MF has been reported to induce ecto-CALR exposure and HMGB1 release in vitro. Its immunogenic potential was confirmed in an in vivo vaccination assay using OVA-expressing EG7 thymoma cells [51]. Along similar lines, Du. et al. reported that this activated derivative of CP induced a robust type I interferon response and vaccinated mice against GL261 glioma cells [52]. Our study showed that induction of DAMPs varied between different NSCLC cell lines after treatment with MF. Although in murine 3LL cells, MF treatment resulted in a significant increase of all DAMPs which was consistent with in vivo findings in which 80% of the mice remained tumor-free.

Only a few studies reported induction of DAMPs after treatment with DOC, a tubulin stabilizing agent, which belongs to the taxane family. The study of Gao et al. showed that DOC treatment significantly upregulated HMGB1 release in human NSCLC cell lines and that HMGB1 levels correlated with improved outcomes in NSCLC patients [53]. Contrarily, Hodge et al. reported that DOC was only able to induce ecto-CALR exposure in human breast, prostate and colorectal carcinoma [54]. In our study, the most consistent treatment among all NSCLC cell lines was DOC, DOC + CARBO and DOC + CDDP.

Finally, we performed a vaccination assay using 3LL cells in an attempt to validate our in vitro and in vivo findings. 3LL cells treated with OXA failed to induce any of the DAMPs, hence it was not surprising that only 20% of the mice vaccinated with OXA were tumor-free. On the other hand, DOC treatment of 3LL cells induced all DAMPs, but only 40% of the mice vaccinated with DOC remained tumor-free. As opposed to DOC alone, combinations with CARBO or CDDP induced all DAMPs in vitro and resulted in 80% and 100% tumor-free mice, respectively. Of note, it is important to mention that although the in vivo vaccination assay remains the gold-standard to confirm ICD up to date, human and murine NSCLC data should be carefully interpreted due to differences between human and murine NSCLC cells in terms of immunogenicity [55].

Whether there is a hierarchical order of importance regarding these DAMPs is not yet clear, although one study assessed the relevance of these DAMPs [56]. Garg et al. blocked each of the DAMPs in vivo and showed that they differ in relevance, based on survival data: the most relevant DAMP was HMGB1, followed by ATP and finally ecto-CALR being the least relevant. Although our in vitro findings demonstrated that DOC induced all DAMPs in 3LL cells, HMGB1 secretion was significantly lower in DOC-treated or OXA-treated cells compared to all other treatments, which might explain why these mice developed a tumor after all.

Recent data on clinical trials of combination strategies with chemotherapy and immunotherapy in NSCLC patients support the idea of including a chemotherapeutic regimen to immunotherapy to mount an effective anti-tumor immune response [57,58]. Thus, obtaining valuable insights into these immunogenic properties paves the way towards more rationally designed combination strategies to optimize clinical outcomes of NSCLC patients.

## 6. Conclusions

In conclusion, with this study we provide novel in vitro and in vivo data on the immunostimulatory effects of clinically relevant chemotherapeutic regimens in NSCLC, with the best results for DOC + CARBO and DOC + CDDP.

## Figures and Tables

**Figure 1 cells-09-01474-f001:**
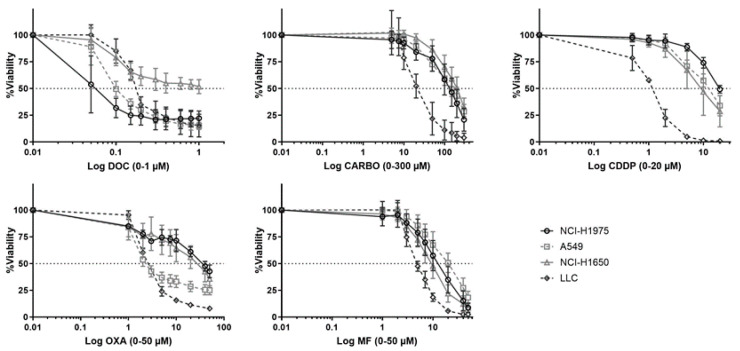
Dose-response curves of different chemotherapeutics in NSCLC cell lines. Survival curve after 72 h of treatment with docetaxel (DOC, 0–1 µM), carboplatin (CARBO, 0–300 µM), cisplatin (CDDP, 0-20 µM), oxaliplatin (OXA, 0–200 µM) and mafosfamide (MF, 0–80 µM) in the NCI-H1975, A549, NCI-H1650 and 3LL cell line. Experiments were performed at least in triplicate. Error bars represent the standard deviation.

**Figure 2 cells-09-01474-f002:**
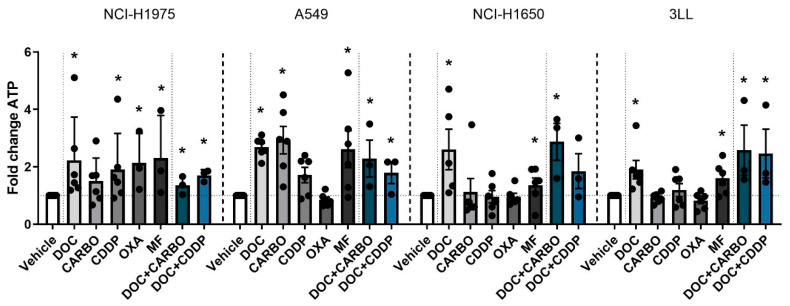
ATP secretion in NSCLC cell lines after treatment with chemotherapy. Early release of ATP was assessed after 24 h of treatment with the IC_50_-72h of docetaxel (DOC), carboplatin (CARBO), cisplatin (CDDP), oxaliplatin (OXA) and mafosfamide (MF) or treatment with the IC_50_-72h of DOC and IC_40_-72h value of either CARBO or CDDP in the NCI-H1975, A549, NCI-H1650 and 3LL cell line. * *p* ≤ 0.05. Error bars represent the standard deviation. Experiments were performed at least in triplicate.

**Figure 3 cells-09-01474-f003:**
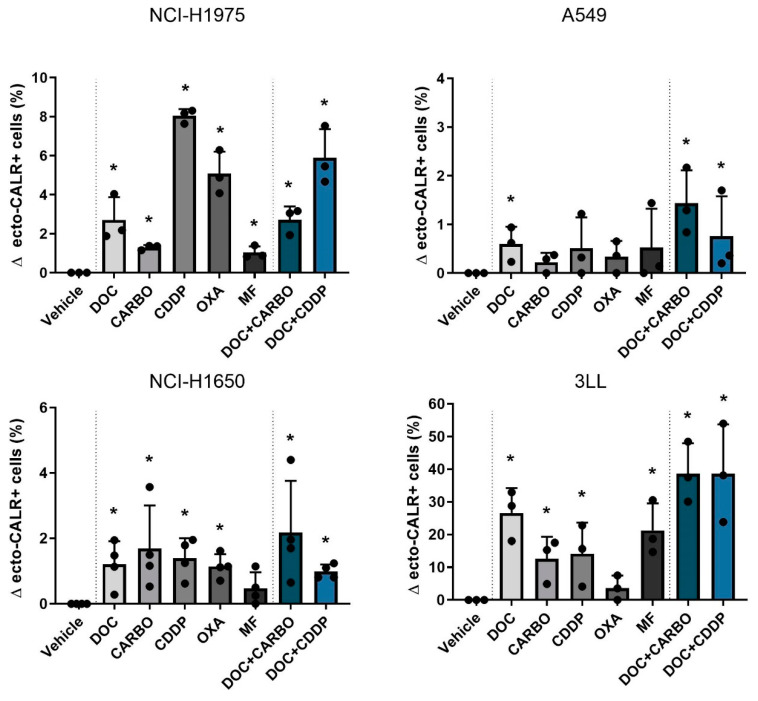
Ecto-CALR exposure in NSCLC cell lines after treatment with chemotherapy. Percentages of ecto-CALR positive (ecto-CALR+) cells were assessed after 48 h of treatment with the IC_50_-72h of docetaxel (DOC), carboplatin (CARBO), cisplatin (CDDP), oxaliplatin (OXA) and mafosfamide (MF) or treatment with the IC_50_-72h of DOC and IC_40_-72h value of either CARBO or CDDP in the NCI-H1975, A549, NCI-H1650 and 3LL cell line. * *p* ≤ 0.05. Error bars represent the standard deviation. Experiments were performed at least in triplicate.

**Figure 4 cells-09-01474-f004:**
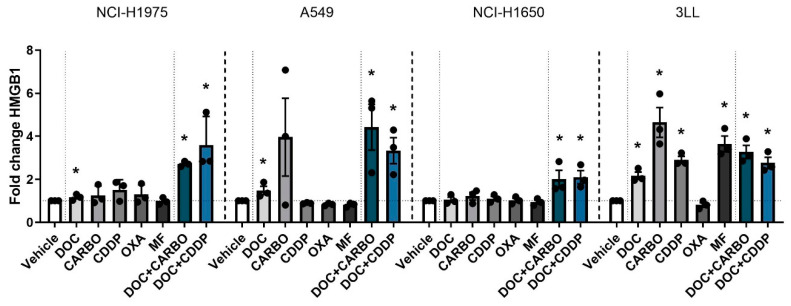
HMGB1 release in NSCLC cell lines after treatment with chemotherapy. Release of HMGB1 (ng/mL) was assessed after 72 h of treatment with the IC_50_-72h of docetaxel (DOC), carboplatin (CARBO), cisplatin (CDDP), oxaliplatin (OXA) and mafosfamide (MF) or treatment with the IC_50_-72h of DOC and IC_40_-72h value of either CARBO or CDDP in the NCI-H1975, A549, NCI-H1650 and 3LL cell line and is depicted as fold change. * *p* ≤ 0.05. Error bars represent the standard deviation. Experiments were performed in triplicate.

**Figure 5 cells-09-01474-f005:**
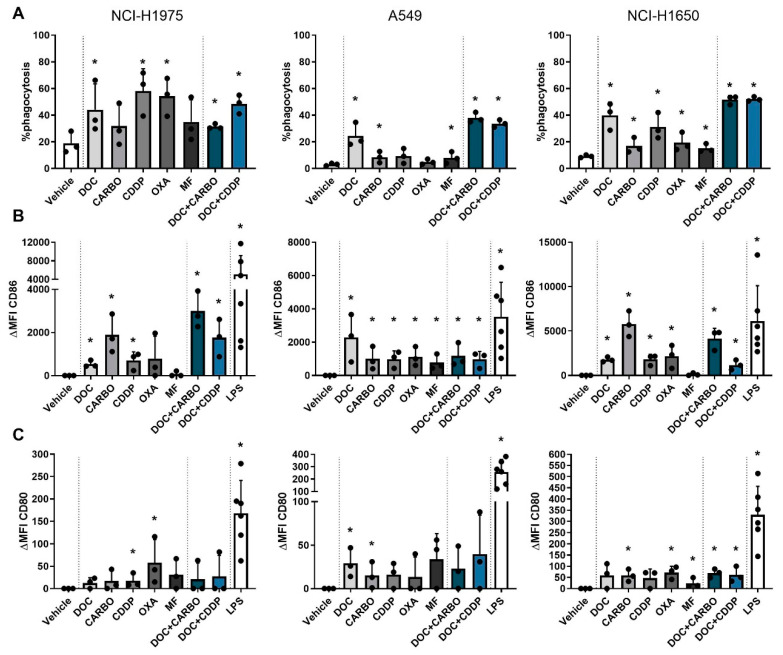
Phagocytic ability and maturation status of DCs in co-culture with different NSCLC cell lines after treatment with chemotherapy. (**A**) Phagocytosis of treated NSCLC cells by DCs and maturation markers (**B**) CD86 and (**C**) CD80 of DCs were assessed after 72 h of treatment with the IC_50_-72h of docetaxel (DOC), carboplatin (CARBO), cisplatin (CDDP), oxaliplatin (OXA) and mafosfamide (MF) or treatment with the IC_50_-72h of DOC and IC_40_-72h value of either CARBO or CDDP in the NCI-H1975, A549 and NCI-H1650 cell line. Data of maturation markers are expressed as delta mean fluorescence intensity (ΔMFI), which was calculated by subtracting the isotype control of each condition and vehicle (PBS), respectively. * *p* ≤ 0.05. Error bars represent the standard deviation. Experiments were performed in triplicate.

**Figure 6 cells-09-01474-f006:**
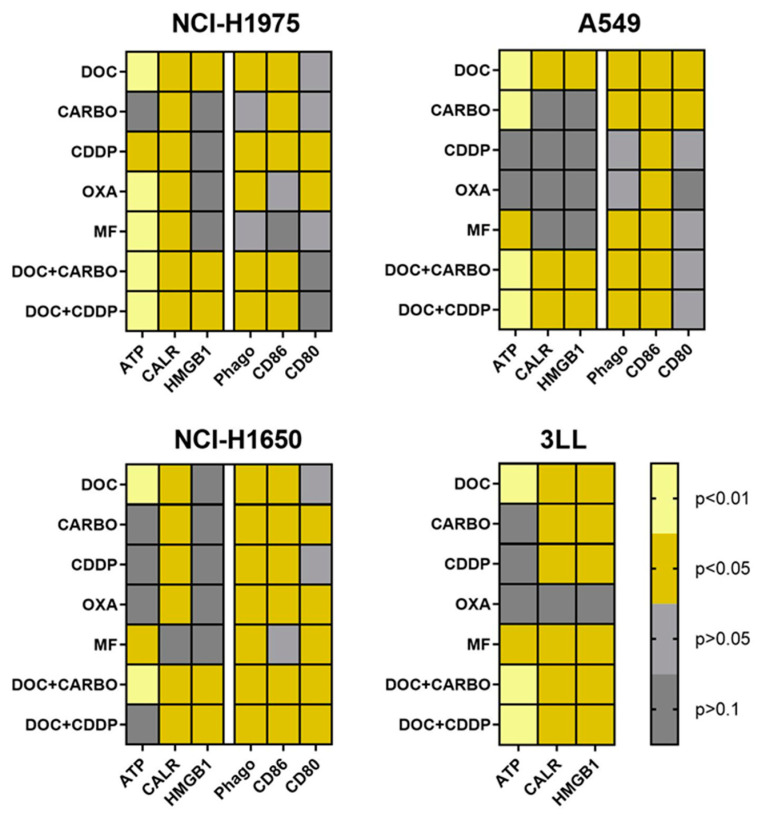
Heatmap overview of DAMPs, phagocytic ability and maturation status of DCs. An overview of the *p*-values of DAMPs is shown for cell lines NCI-H1975, A549, NCI-H1650 and 3LL. Phagocytic ability and maturation status of DCs is shown for cell lines NCI-H1975, A549 and NCI-H1650. Light yellow: *p* ≤ 0.01; dark yellow: *p* ≤ 0.05 (if increased compared to vehicle); light grey: *p* > 0.05; dark grey: *p* > 0.1.

**Figure 7 cells-09-01474-f007:**
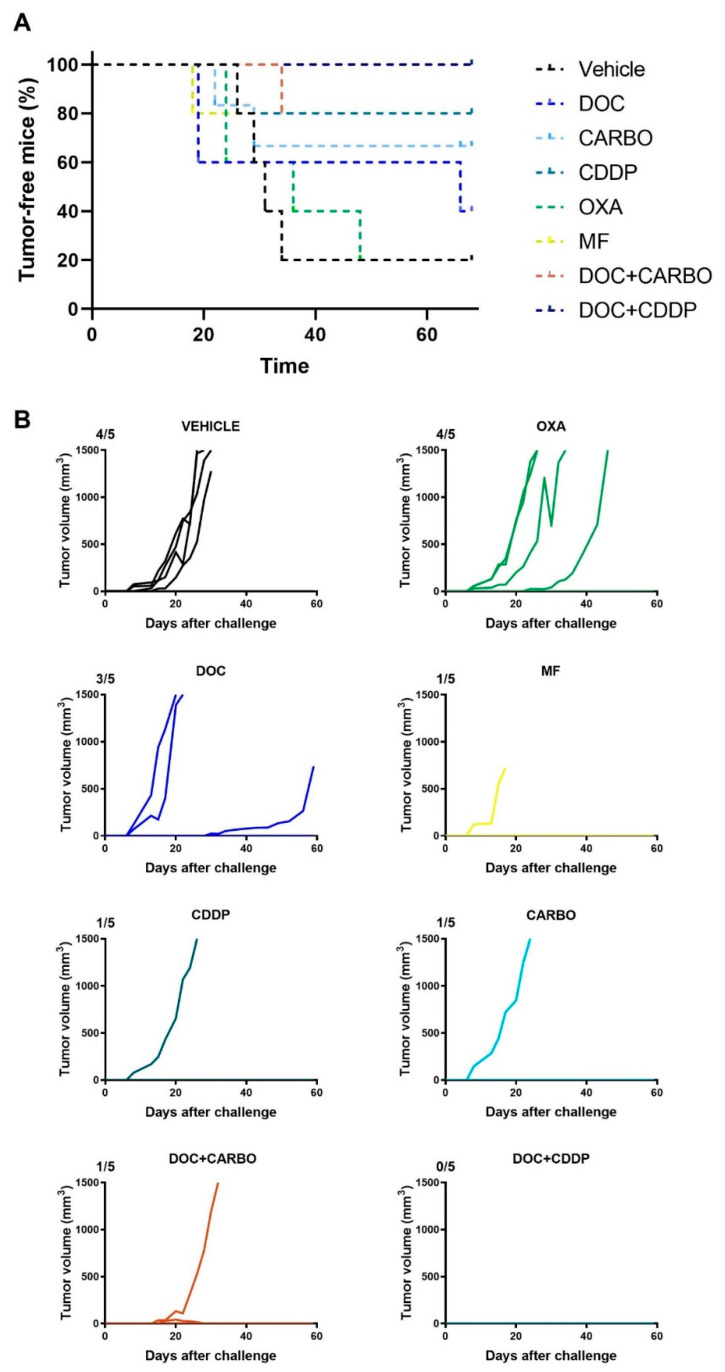
In vivo vaccination underscores immunogenic capacity of clinically relevant chemotherapeutics in the 3LL mouse model. (**A**) Percentage (%) of tumor-free mice and (**B**) follow-up of tumor growth (mm^3^) of C57BL/6J mice for 60 days or until endpoint was reached (volume ≥ 1500 mm^3^). Five mice were assigned to each condition: Vehicle (PBS), docetaxel (DOC), carboplatin (CARBO), cisplatin (CDDP), oxaliplatin (OXA) and mafosfamide (MF), DOC + CARBO or DOC + CDDP. Ratios on top of each graph represent the number of mice that developed a tumor during the follow-up period.

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
