# Peer review of "Clinically Relevant Chemotherapeutics Have the Ability to Induce Immunogenic Cell Death in Non-Small Cell Lung Cancer"

_cells, 2020, doi:10.3390/cells9061474_

Round 1

Reviewer 1 Report

The manuscript entitled "Clinically relevant chemotherapeutics have the ability to induce immunogenic cell death in non-small cell lung cancer" demonstrated that chemotherapeutic regimens in NSCLC patients may have the ability to induce immunogenic cell death.

Minor comments:

  • The Authors should control all the acronym through the text and provide the extensive form when they first appear. 
  • In the Discussion section "The ability of chemotherapy to elicit ICD has become a thoroughly investigated phenomenon,in both in vitro and in vivo settings.", the Authors should provide a reference for this statement.

Author Response

Point 1: The Authors should control all the acronym through the text and provide the extensive form when they first appear.

Response 1: Thank you for noticing these inconsistencies. We have thoroughly screened the text and adjusted the form where needed using the ‘Track Changes’ function.

Line 44: Food and Drug Administration (FDA) and European Medicines Agency (EMA)
Line 68: adenosine triphosphate (ATP)
Line 91: Dulbecco's Modified Eagle Medium (DMEM)
Line 94: Roswell Park Memorial Institute Medium (RPMI)
Line 140: cluster of differentiation (CD) 14 positive (+) cells
Line 151: Supernatant (SN)
Line 166: vehicle (phosphate-buffered saline; PBS)
Line 332-333: interferon-β (IFN-β), tumor necrosis factor-α (TNF-α)
Line 334: SN instead of supernatant

Point 2: In the Discussion section "The ability of chemotherapy to elicit ICD has become a thoroughly investigated phenomenon, in both in vitro and in vivo settings.", the Authors should provide a reference for this statement.

Response 2: We have included a reference for this statement (line 384).

Reviewer 2 Report

The manuscript demonstrates the clinically applicable chemotherapies having ability to induce immunogenic cell death in non-small cell lung cancer. Non-small cell lung cancer is the most deadly form of lung cancer with 15% survival rate in patients within 5 years with no effective treatment options. The authors have evaluated 5 chemotherapeutic drugs viz. docetaxel, carboplatin, cisplatin, oxaliplatin and mafosfamide for their ability to induce immunogenic cell death in NSCLC cell lines. The authors have use various assays to determine the immunogenicity of drugs and their combinations. The study also tested them in preclinical model of NSCLC. Overall, the concept is intriguing and the studies are well performed, the results support the conclusions.

A suggestion to improve the quality of the manuscript is that the figures 2-5 generated are very generic and describe the effect of individual treatment as well as drug combination. The authors should attempt to represent these figures in different format.

Author Response

Point 1: A suggestion to improve the quality of the manuscript is that the figures 2-5 generated are very generic and describe the effect of individual treatment as well as drug combination. The authors should attempt to represent these figures in a different format.

Response 1: We would like to thank the reviewer for reading the manuscript and his/her valuable remark. We have adjusted Figure 2-5 to have a less generic representation of the data.